Premature feather loss among common tern chicks in Ontario: the return of an enigmatic developmental anomaly

Arnold Jennifer M. 1 jma25@psu.edu
Tyerman Donald J. 2
Crump Doug 3
Williams Kim L. 3
Oswald Stephen A. 1
1 Division of Science, Pennsylvania State University, Berks Campus , Reading, PA , United States
2 Presqu’ile Provincial Park, Ontario Parks , Brighton, Ontario , Canada
3 National Wildlife Research Centre, Environment Canada , Ottawa, Ontario , Canada
Boersma P. Dee
Electronic publication date: 2016 May 18
Publication date: 2016
Volume: 4
Electronic Location ID: e1959
Received 2015 Dec 3; Accepted 2016 Apr 1
Copyright: ©2016 Arnold et al.
Copyright year: 2016
Copyright holder: Arnold et al.
License: This is an open access article distributed under the terms of the Creative Commons Attribution License, which permits unrestricted use, distribution, reproduction and adaptation in any medium and for any purpose provided that it is properly attributed. For attribution, the original author(s), title, publication source (PeerJ) and either DOI or URL of the article must be cited.
License URL: https://creativecommons.org/licenses/by/4.0/

Keywords: Contaminants, Feather loss, Nutritional stress, Premature moult, Avian virus, Pathogens, Induced moult, Algal toxins, Great Lakes

Funding: Pennsylvania State University Berks Campus Ned Smith Center for Nature and Art Friends of Presqu’ile Provincial Park Baird Ornithological Club Funding for this study was provided by the Pennsylvania State University Berks Campus, Ned Smith Center for Nature and Art, Friends of Presqu’ile Provincial Park and Baird Ornithological Club. The funders had no role in study design, data collection and analysis, decision to publish, or preparation of the manuscript.

==============================
In July 2014, we observed premature feather loss (PFL) among non-sibling, common tern Sterna hirundo chicks between two and four weeks of age at Gull Island in northern Lake Ontario, Canada. Rarely observed in wild birds, to our knowledge PFL has not been recorded in terns since 1974, despite the subsequent banding of hundreds of thousands of tern chicks across North America alone. The prevalence, 5% of chicks (9/167), and extent of feather loss we report is more extreme than in previous reports for common terns but was not accompanied by other aberrant developmental or physical deformities. Complete feather loss from all body areas (wing, tail, head and body) occurred over a period of a few days but all affected chicks appeared vigorous and quickly began to grow replacement feathers. All but one chick (recovered dead and submitted for post-mortem) most likely fledged 10–20 days after normal fledging age. We found no evidence of feather dystrophy or concurrent developmental abnormalities unusual among affected chicks. Thus, the PFL we observed among common terns in 2014 was largely of unknown origin. There was striking temporal association between the onset of PFL and persistent strong southwesterly winds that caused extensive mixing of near-shore surface water with cool, deep lake waters. One hypothesis is that PFL may have been caused by unidentified pathogens or toxins welling up from these deep waters along the shoreline but current data are insufficient to test this. PFL was not observed among common terns at Gull Island in 2015, although we did observe similar feather loss in a herring gull Larus argentatus chick in that year. Comparison with sporadic records of PFL in other seabirds suggests that PFL may be a rare, but non-specific, response to a range of potential stressors. PFL is now known for gulls, penguins and terns.

Introduction

In common terns, Sterna hirundo, the first feather molt occurs between four and seven months after fledging (Nisbet, 2002). Thus, feather loss for chicks prior to fledging is not part of normal development (Wails, Oswald & Arnold, 2014). In July 2014, we observed premature feather loss (PFL) among pre-fledging common tern chicks at Gull Island in northern Lake Ontario, Canada. This condition, whereby developing chicks lose their wing, tail, head and body feathers, has rarely been documented in wild birds. To our knowledge, PFL has only been reported for terns in coastal, eastern North America and at Indian Ocean breeding islands between 1970 and 1974 (Gochfeld, 1971; Gochfeld, 1975; Hays & Risebrough, 1972; Feare, 1974). At that time, it was cautiously associated with contaminant burdens, chiefly mercury and polychlorinated biphenyls (PCBs) (Hays & Risebrough, 1972; Gochfeld, 1980), and pathogenic organisms and their toxins (Bourne, Bogan & Bullock, 1977). However, researchers were unable to rule out the possibility of other stressors (such as trauma, cancers, allergens, infections, and genetic factors) (Gochfeld, 1971).

In general, feather loss (aside from molt) can be caused by starvation, nutritional deficiencies, pecking, shock molt, and Psittacine Beak and Feather Disease (Spearman, 1980; Hughes, 1985; Leeson & Walsh, 2004; Møller, Nielsen & Erritzoe, 2006). Although commonly reported in poultry (Hughes, 1985; Leeson & Walsh, 2004), feather loss is rarely reported in wild birds except for some pathogenic infections (e.g., in psittacines; Ha et al., 2007; but see Table 1), and occasional observations from pecking during territorial disputes in colonial birds (e.g., Nisbet, Wilson & Broad, 1978). In North America, PFL in common terns was reported locally in the vicinity of Long Island, New York, as well as a few locations in coastal Connecticut and Massachusetts, between 1970 and 1974 (Hays & Risebrough, 1972; Nisbet, 1972; Gochfeld, 1980). However, to the best of our knowledge, and despite the banding of over 850,000 common terns in North America alone since then (D Bystrak, pers. comm., 2016), no further cases have been reported anywhere.

Table 1 Reports of premature feather loss in wild populations of waterbirds.

In all reports, feather-loss in some individuals studied was extensive.

Species	Location	Years	Explanation	Source	
Common tern (Sterna hirundo)	Long Island, NY	1970–1974	Proposed link to contaminants (e.g., PCBs, mercury)	Gochfeld (1971), Gochfeld (1975) and Hays & Risebrough (1972)	
Sooty tern (Sterna fuscata)	Seychelles	1973	Tick-borne soldado-virus	Feare (1974)	
Greater Black-backed gull (Larus marinus)	Witless Bay, Newfoundland	1984	Unknown cause	Roy, Threlfall & Wheeler (1986)	
Herring gull (Larus argentatus)	Witless Bay, Newfoundland	1984	Unknown cause	Roy, Threlfall & Wheeler (1986)	
African penguin (Spheniscus demersus)	South Africa	1989	Malnutrition	Van Heezik & Seddon (1992)	
Emperor penguin (Aptenodytes forsteri)	Cape Washington, Antarctica	mid-1990s	Unknown cause	Reported in Varsani et al. (2014)	
Magellanic penguin (Spheniscus magellanicus)	Argentina	2007–2008	Unknown cause	Kane et al. (2010)	
African penguin (Spheniscus demersus)	South Africa	2008	Unknown cause	Kane et al. (2010)	
Adelie penguin (Pygoscelis adeliae)	Cape Crozier, Antarctica	2011	Unknown cause	Reported in Varsani et al. (2014)	
Common tern (Sterna hirundo)	Gull Island, Lake Ontario	2014	Unknown cause, possible link to environmental conditions	This study	
Adelie penguin (Pygoscelis adeliae)	Ross Is. & Antarctic Peninsula	2011–2015	Proposed avian virus (polyomavirus, novel astrovirus)	Barbosa et al. (2015) and Grimaldi et al. (2015)	
Herring gull (Larus marinus)	Gull Island, Lake Ontario	2015	Unknown cause	J Arnold & S Oswald (2015, unpubl. obs.)	

Our detection of PFL in common terns is just one of an increasing number of reports of this condition in a range of wild bird species (Table 1). Given that this condition is highly visible, the recent escalation in its detection worldwide may indicate changing health risks for birds and other wildlife. Here, we describe the chronology and progression of PFL at Gull Island, Ontario in 2014 and review associated evidence to narrow down potential causes of this aberrant development. We aim to increase awareness of this phenomenon among ornithologists and the wider scientific community to ensure timely reporting of future occurrences that may lead to a better understanding of the underlying causes of PFL.

Materials & Methods

Since 2008 we have studied the reproductive biology of common terns annually at Gull Island, Presqu’ile Provincial Park, ON, Canada (43°59′N, 77°45′W) under appropriate authorizations: Canadian Wildlife Service Permits CA 0218, CA 0234, CA 0242, CA 0267, CA 0308; Environment Canada Bird Banding Permits 10431 V and 10431 W; Ontario Parks Authorizations to Conduct Research in a Provincial Park 2008–2015 inclusive; and Institutional Animal Care and Use Committee (IACUC) of Pennsylvania State University approved protocols 45332, 36295, 28103. Following standard protocols (Arnold, Nisbet & Oswald, 2016), each year nearly all chicks were banded within 1–2 d of hatching and chicks from all study nests (∼80–130) were recaptured and weighed regularly (every ∼1–4 d) from hatching until fledging. Number of chicks banded in each year varied between 108 and 297 depending on the intensity of nest predation by night herons and gulls (unrelated to our monitoring). In 2014, chicks were recaptured every ∼1–4 d until 24 July (when the majority of chicks were 28 d or older; median fledging age is 24–25 d at this colony) and then weekly until 20 August.

On discovery of premature feather loss (PFL) in multiple individual chicks (that previously exhibited normal feathering) at Gull Island, we took a range of photographs (head, tail, wing, body) and also measured wing length (maximum wing chord) and tail length (maximum length of longest outer tail feather) at each subsequent recapture using a 300 mm wing rule. The same measurements were made for a sample of 7 normally developing tern chicks during this time period; mass measurements were available from 159 normally developing chicks. The carcass of one subsequently dead PFL chick was sent to D Campbell, of the Canadian Wildlife Health Cooperative, Ontario Veterinary College for post-mortem.

Repeated mass measurements from eight chicks that exhibited PFL were used to construct a composite growth curve for comparison to larger samples of normally developing chicks. Fledging masses were estimated for 29 normally-feathered chicks as their mass on the date of last recapture prior to fledging (median fledging age = 24–25 d) and compared to the masses of 7 PFL chicks at their last recapture (age range: 20–41 d, median = 27–28 d) using a Student’s t-test. Median wing and tail growth rates (slopes of measurements within individuals) for chicks exhibiting PFL and normally-feathered chicks were compared using Mann Whitney U-tests. For these analyses, only chicks with similar hatch dates and multiple measurements between ages 20 d to 36 d were included in analyses (6 PFL chicks and 3 normally-feathered chicks). For each PFL chick, we used current wing length and the calculated rate of daily wing growth (slope of maximum chord) on the last day it was measured (range 16–24th July) to project its fledging date (the date at which wing lengths would have equaled 180 mm; the smallest wing length for a normal, fledged common tern chick in 2014). The assumption of a constant growth rate between ∼10 d of age and fledging is supported by data on wing growth in this species (LeCroy & Collins, 1972).

Local, hourly weather data from the Trenton, ON, weather station (44°7′N, 77°32′W; 21 km to the northwest) were downloaded from Environment Canada (http://climate.weather.gc.ca). From these we calculated daily (24 h) means from 1 May to 31 July 2014 for air temperature, relative humidity, wind speed, wind direction, visibility and standard atmospheric pressure, and also minimum nighttime temperatures and maximum daytime air temperatures. Lake surface temperature was retrieved from Environment Canada (http://weather.gc.ca) for the nearest, near-shore, weather buoy (Ajax, ON, Station 45159, 43°46′N 78°59′W, 105 km E of Gull Island). For comparative purposes, the same data were also extracted for all years in which data coverage was sufficiently comprehensive (2009–2015).

Analyses were conducted in R (R Core Team, 2015). All means are reported with ±1 SD, all medians with quartiles [upper, lower].

Figure 1 Plumage characteristics resulting from of premature feather loss (PFL) in common tern chicks at Gull Island in 2014 (A, C, E) versus normal development (B, D, F; overhead photo in B is taken from the Common Tern Aging Guide: Wails, Oswald & Arnold, 2014).

In each case, whole body (A, B), wing (C, D) and tail (E, F) are shown (pictures taken between 9 and 18 July). Chicks shown are between 21 and 27 d of age (fledging usually occurs between 21 and 29 days; Nisbet, 2002).

Figure 2 Growth in mass of the eight chicks exhibiting premature feather loss [PFL] (black lines = 95% confidence intervals) superimposed over the range of mass development for normal chicks in 2014 (n = 159 chicks, grey shading = area between 95% confidence intervals).

For PFL chicks measured later in development (>30 d of age), when fewer measurements were available, individual data points are plotted.

Results

Premature feather loss (PFL) was first discovered in two non-sibling, common tern chicks at Gull Island on 5 July, 2014. Initial symptoms that we recorded at this time were missing feathers on the head and body, similar in extent to that sometimes resulting from territorial aggression by neighboring adults or chicks (but more extreme and without the associated laceration, bruising or hemorrhaging). However, by 8 July these two chicks had lost down, primaries and most feathers from all areas of the head, body, tail and wings, and in one chick, pin feathers were already growing back in places (Fig. 1). On this same day, two other chicks exhibited PFL symptoms for the first time. Three chicks were found with PFL between 9 and 11 July and another one chick exhibited PFL on 24 July. We also discovered an unbanded, medium-sized chick (estimated as 13–15 d of age) exhibiting PFL on 15 August (which is excluded from the analyses that follow). Thus a total of nine chicks were found that exhibited PFL out of the 167 handled and banded (5.3%) at Gull Island in 2014. Although all eight of the banded PFL chicks were from either 2- or 3-chick broods, none had siblings that showed signs of PFL.

No evidence of feather dystrophy was found in the dead PFL chick sent for post-mortem analysis, reducing the likelihood that this PFL resulted from a viral infection (such as that from Circovirus and Papovavirus) (D Campbell, pers. comm., 2015). It also showed no obvious signs of infection or abnormality in heart, lung, liver or kidney tissues (D Campbell, pers. comm., 2015). There was a high level of cellularity in the pulp of the feathers, but the significance of this is unclear (D Campbell, pers. comm., 2015).

The mean age of chicks when first exhibiting PFL was 18 (±3.7) d (range: 15–26 d). Visual comparison between normal plumage development and that of chicks exhibiting PFL is provided in Fig. 1. Although growth in mass for chicks exhibiting PFL appeared normal initially, after 10 d of age ∼88% lost 11–20% of body mass (3–9d prior to PFL) or had slower than average growth rates. Reduced mass persisted until normal fledging age (∼25 d) (Fig. 2). Although median rates of both wing and tail growth were slightly higher for PFL chicks than normally-feathered chicks at similar ages, these differences were not statistically significant (wing: PFL = 5.5 [4.3, 6.2] mm/d (n = 6), Normal = 4.0 [2.8, 5.3] mm/d (n = 3), W = 5, P = 0.38; tail: PFL = 3.8 [2.9, 4.9] mm/d (n = 6), Normal = 3.3 [2.2, 3.5] mm/d (n = 3), W = 3, P = 0.17).

The seven surviving banded chicks that exhibited PFL in July were last seen between 21 and 42 d of age (mean: 29.1 ± 6.5 d). Mean body mass at their last measurement date (between 16th and 24th July) was 119.3 (±9.2) g, effectively identical to the colony average (118.3 ± 8.1 g, n = 29; t34 = 0.3, P = 0.77) for normally-feathered birds of similar age (the normal fledging age median: 24–25 d). Mean wing length of the PFL chicks was 125 (±23) mm during this same time period (compared to a minimum of 180 mm for a normally-feathered, fledged chick in 2014). The earliest and latest projected fledging dates for the PFL chicks were 26 July and 20 Aug (mean: 2 Aug ± 9 d), respectively. During a visit to the colony on 11 Aug, we estimated there to be between 15 and 25 active common tern broods (based on counts of persistently dive-bombing adults) of the 107 total nests that were initiated in 2014.

Figure 3 Changes in mean daily weather conditions relative to hatching and exhibiting premature feather loss.

Changes in mean daily weather conditions, (A) near-shore lake surface temperature, (B) air temperature, (C) wind speed, and (D) wind direction (maximum gust), and correspondence with distribution of hatching dates (blue boxplots and outlier) and dates of first exhibiting premature feather loss [PFL] (green boxplots and outlier). Trend lines are 7-day running average of the weather variable. Grey shading highlights the period of plummeting near-shore surface water temperatures (2–10 July).

The period in which PFL was first detected in common tern chicks (and three days immediately prior to it: 2–10 July) was characterized by plummeting near-shore, lake surface temperatures (Fig. 3A) and falling air temperatures (Fig. 3B) as well as stronger southwesterly winds (Figs. 3C and 3D) and rising atmospheric pressure (not shown). Minimum nighttime temperatures and maximum daytime temperatures were highly correlated with mean air temperature (r89 = 0.82 and r90 = 0.99, respectively) and showed the same response (not shown). However, there were no obvious changes in relative humidity or visibility at this time (also not shown). The coincidence of these weather conditions with the chick-rearing schedule of common terns was unusual: across the seven years for which data were available, only one other similar event (July 20th 2013) was detected, although in this case lake surface temperature change was less extreme (8 C°vs. 13 C°) and was primarily a result of one, strong, north-westerly storm rather than persistent south-westerlies.

Discussion

The premature feather loss (PFL) that we observed in common tern chicks at Gull Island in 2014 is similar in two ways to that described by researchers working on the Atlantic coast of North America in the early 1970s. Firstly, we only observed it in chicks when they were between 2 and 4 weeks of age, the same age as noted by Hays & Risebrough (1972) and similar to Gochfeld (1971) (3–5 weeks), although in this latter case PFL was mostly restricted to chicks closer to normal fledging age (Gochfeld, 1971, M Gochfeld, pers. comm., 2015). Secondly, in all reported cases, a proportion of chicks that shed feathers grew replacements and appeared otherwise healthy and vigorous (Gochfeld, 1971; Hays & Risebrough, 1972). This was true of nearly all affected chicks at Gull Island in 2014. In fact, our PFL chicks, although they consequently had shorter wings and tails than normally-feathered chicks of the same age (Fig. 1), showed no reduction in feather growth rates following feather loss. This is unsurprising, since feather growth appears highly conserved (e.g., under conditions of nutritional stress; Bize, Metcalfe & Roulin, 2006; Lyons & Roby, 2011).

There are a few interesting differences between the PFL we observed and that previously documented both in terns and other waterbirds (Table 1). Complete feather loss (Fig. 1) is the extreme among terns in previous reports, as some birds only lost primaries and/or tail feathers (Gochfeld, 1971; Gochfeld, 1975; Hays & Risebrough, 1972; Feare, 1974; Bourne, Bogan & Bullock, 1977). Similar complete feather loss and regrowth has been observed in other species, e.g., greater black-backed gulls (Larus marinus) (Roy, Threlfall & Wheeler, 1986), and African (Spheniscus demersus) and Magellanic (S. magellanicus) penguins (Kane et al., 2010). The incidence of PFL at our colony in 2014 was 5% (9/167) of all chicks, higher than that reported previously for common terns (0.5–1.1%; Hays & Risebrough, 1972; Gochfeld, 1975), although similar rates have been noted for other terns and gulls (Feare, 1974; Roy, Threlfall & Wheeler, 1986) and even higher incidences in rehabilitated penguins (Kane et al., 2010). Unlike all other reports for terns, we did not observe concurrent developmental abnormalities (e.g., crossed-bills, aberrant limb development; Feare, 1974; Gochfeld, 1975; Bourne, Bogan & Bullock, 1977) in PFL chicks or any other common tern chicks at our site. Lack of concurrent abnormalities is, however, consistent with historical records for gulls (Roy, Threlfall & Wheeler, 1986) and contemporary reports among penguins (Kane et al., 2010; Barbosa et al., 2015; Grimaldi et al., 2015). Colony-wide hatching success also did not appear any different from in previous years (J Arnold & S Oswald, 2014, unpublished data), suggesting an absence of gross embryonic deformity (such situations were previously linked with PFL; Hays & Risebrough, 1972). Interestingly, previous studies in the lower Great Lakes between 1971 and 1973 detected a high prevalence of deformity among common tern chicks (including one chick at Presqu’ile Provincial Park) but no cases of PFL (Gilbertson, Morris & Hunter, 1976).

While growth rates of wings and tails of common tern chicks recovering from PFL in 2014 appeared normal, PFL chicks had lower masses than normally-feathered chicks between approximately 10 d of age and normal fledging age (∼25 d). Slowed growth and/or acute weight loss prior to feather loss may indicate the onset or trigger of this condition in some cases. After PFL the maximum weight differences observed (∼30 g on average at 18 d of age; Fig. 2) likely exceeded pre-PFL weight loss (7–19 g) plus the weight of lost plumage (5–9 g, based on estimates of plumage mass for similar species: 5–7% of body mass; Braune & Gaskin, 1987; Marks, 1993). As chicks recovering from PFL appeared, vigorous and highly aggressive in some cases, this later weight difference is unlikely to result from a competitive disadvantage during provisioning events (e.g., Oswald et al., 2012) but, in part, from a preferential channeling of energy to feather regrowth. Such a relationship between PFL and growth has been previously documented for penguins (Kane et al., 2010) but not terns (Gochfeld, 1971). PFL did not appear to affect the known tendency for tern chicks to overshoot adult mass and subsequently lose mass at this site, as the proportion of PFL chicks exhibiting this growth trajectory (25% (2/8 chicks)) was similar to that previously reported (34–38%; Arnold, Nisbet & Oswald, 2016).

Other reports of PFL in terns have been for chicks late in the breeding season (Gochfeld, 1971) but at Gull Island in 2014, PFL was first observed in early July (median hatching date across all chicks in 2014 was 21 June) and an active colony persisted well into mid-August. There was a striking association between the timing of the onset of PFL at Gull Island in 2014 and strong, southwesterly winds, falling air temperatures, and plummeting (up to 13 C°) near-shore, lake surface temperatures (daily means shown in Fig. 3): conditions characteristic of mixing of cold, deep lake waters with warmer surface waters. The extent of lake-surface temperature change observed in 2014 is rare, comparable to only one other weather event within the last seven years at a similar stage in the common tern breeding cycle (July 2013). However, this 2013 weather event was driven by strong, northwesterly winds and showed less extreme temperature change; presumably more northerly winds gave a lower potential for mixing of deep lake waters with shallow coastal waters of the northern lake shore (where Gull Island is located). There was also no evidence of PFL among developing chicks in 2013. Therefore, one hypothesis is that the PFL we observed in 2014 may have been caused by unidentified pathogens or toxins welling up from deep lake waters, but current data are insufficient to test this.

Exposure to as-yet-unknown toxins or pathogens seems the most likely current explanation of PFL in common tern chicks at Gull Island. Most affected chicks were between 1–2 weeks of age during the time period when PFL was first observed (5 July), although one PFL chick was hatched at this time and the egg of another chick that developed PFL in August was laid at this time. We favor exposure to unknown toxins or pathogenic organisms as the causal mechanism.

Comparison with sporadic records of PFL in other birds (Table 1) suggests that PFL may be a rare, but non-specific response to a range of potential stressors, including environmental contaminants, viral and bacterial infections, tick-borne disease or nutritional deficiency. Given that PFL is very obvious in affected chicks (Fig. 1) but is seldom reported despite the great number of studies that should detect it, the low incidence of reporting likely represents a rare condition rather than simply underreporting (Gochfeld, 1971; Roy, Threlfall & Wheeler, 1986; Barbosa et al., 2015; Grimaldi et al., 2015). In 2015, we found no evidence of PFL in common terns despite similar nesting numbers and research protocols, but we did observe a single herring gull (L. argentatus) chick exhibiting premature feather loss (J Arnold & S Oswald, 2015, unpubl. obs.). Thus, the recent records of PFL in penguins (including adults; Grimaldi et al., 2015), gull and tern chicks may perhaps indicate widespread environmental changes that could lead to health risks for birds and other wildlife, especially if pathogenic organisms are responsible (Bourne, Bogan & Bullock, 1977).

Supplemental Information

Data S1 Raw data underlying results in the manuscript

Contains the following:Summaries of weather data downloaded from Environment Canada (see paper and detailed links for more details about each metric); data for measurements of mass and wing for PFL chicks and mass for normally developing chicks at Gull Island in 2014; daily wing (maximum wing chord) growth rates for PFL chicks and matched controls; daily tail (maximum length of longest outer tail feather) growth rates for PFL chicks and matched controls ;Measurements of mass (g) of chicks different ages (days) through development used in construction of composite growth curves from 8 PFL chicks and 159 control chicks at Gull Island in 2014; notes taken on the condition of chicks and the progression of PFL for the eight PFL chicks.

Click here for additional data file.

We thank S Smith and our field assistants from 2014: M Bush, G Care, S Conway, J Kenna, P McFarland Jr, L Post, K Ringler, and J Smith, as well as J Edwards (2015 assistant). We are greatly indebted to the following for feedback on earlier drafts of the manuscript and/or our observation of PFL: ICT Nisbet, M Gochfeld, CM Custer, I Barker, R Shirley, OJ Kane, T Custer, D Campbell, RD Morris, JM Levengood, J Farquhar, C Grooms, L Harper, DV Weseloh, FJ Cuthbert, C Adams, M LaBarr, and CN Wails. We are also grateful to Olivia Woods and two anonymous reviewers for helpful comments that improved the original manuscript.

Additional Information and Declarations

Competing Interests

Author Contributions

Animal Ethics

Field Study Permissions

Data Availability

Donald J. Tyerman is an employee of Ontario Parks at Presqu’ile Provincial Park. Doug Crump and Kim L. Williams are employees of the National Wildlife Research Centre in Ottawa.

Jennifer M. Arnold and Stephen A. Oswald conceived and designed the experiments, performed the experiments, analyzed the data, contributed reagents/materials/analysis tools, wrote the paper, prepared figures and/or tables, reviewed drafts of the paper.

Donald J. Tyerman performed the experiments, reviewed drafts of the paper.

Doug Crump and Kim L. Williams analyzed the data, contributed reagents/materials/ analysis tools, prepared figures and/or tables, reviewed drafts of the paper.

The following information was supplied relating to ethical approvals (i.e., approving body and any reference numbers):

The Institutional Animal Care and Use Committee (IACUC) of Pennsylvania State University. Protocols: 45332, 36295, 28103.

The following information was supplied relating to field study approvals (i.e., approving body and any reference numbers):

Canadian Wildlife Service Permits CA 0218, CA 0234, CA 0242, CA 0267, CA 0308; Environment Canada Bird Banding Permit 10431 V and 10431 W.

The following information was supplied regarding data availability:

Raw data is provided as Data S1.

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
