# Peer review of "Premature feather loss among common tern chicks in Ontario: the return of an enigmatic developmental anomaly"

_PeerJ, doi:10.7717/peerj.1959_

## Round 0.1 · original submission · Major Revisions

Reviewers thought it was important to get reports of feather loss in wild birds into the literature, but the analyses seem to be a bit of a stretch and the manuscript generally lacks context for the results. In particular, three areas need clarifying and/or additional information. 1) The corticosterone analysis is not framed correctly, as feathers being regrown may not contain corticosterone from before the feather loss. Questions about the methods used to analyze corticosterone need to be addressed. 2) The environmental analysis lacks long-term context. Have these wind and temperature conditions occurred in other years of the study without feather loss? How unusual are the conditions over the last decade? 3) A discussion of common tern chick molt schedule is needed to give the feather loss context.

·

Basic reporting

-This paper provides evidence that the feather loss disorder first observed in 2006 continues to emerge in new species and that the cause remains unknown. While the manuscript needs revisions, the message is important and needed to further the understanding of this rare but emerging disorder.
- Need more background on Corticosterone. Why was it tested? What can it tell us about the cause of feather loss?

Experimental design

- Need more specifics (See General Comments to the Author).
- Need to look at all study years' weather data before making claim that 2014 event was so aberrant that it may have caused feather loss.
- Need to clarify how feathers grow in terns and in turn how cotricosterone is stored in feathers.

Validity of the findings

- Need to steer paper message away from "re-emergence". The previous feather loss reports had other symptoms associated with the disorder while recent feather loss reports mirror the feather loss reported in this paper.
- Message of paper. Common terns can be added to the burgeoning list of species that are affected by a feather loss disorder that was first observed in African Penguins in 2006. The disorder is rare, does not appear to impact mortality, and may be related to extreme weather conditions occurring during chick rearing.

Additional comments

General Comments
• Need to be consistent with naming chick groups. Don’t use control. Use something more descriptive, for example, normally-feathered.

Title
• Remove “Contemporary” from the title
• Remove “revisted” from short title. Try “Premature feather loss in terns”. Revisted seems like your doing a new assessment of previous feather loss cases, while you mention other cases the objective of your paper is to say that it has also been seen in terns.

Abstract
• Line 29: Change “…5% prevalence…” to “…5% (9/167) prevalence…”
• Line 35: “…including shed feathers and regrowing…” statement is misleading. I took this to mean you took feathers from each bird as it was losing feathers and again from the same bird as it grew in new feathers.
• Line 37: Remove sentence about weather. Too grand of a claim with little backing. How anomolous is the strong South-westerly winds? Has it occurred during this same time in other years and no PFL chicks were found?

Introduction
• Line 53: Move mention of domestic poultry exclusively to 3rd paragraph. Mentioning it here makes me want to know what causes it in domestic poultry which you don’t explain until paragraph 3.
• Line 61: Move citations after “… in two main capacities”.
• Lines 61-63: Reword sentence. Split into two sentences. Something like… Colonial seabirds and waterbirds are recognized as indicators of aquatic environments (Citations). Changes in seabird and waterbird demographics or in the organisms physiology may indicate changes in the aquatic environment.
• Line 63-68: Needs to be reworded. Message is unclear.
• Line 64: What is an endpoint?
• Paragraph 61-70: I like the point that seabirds are indicators of aquatic environmental conditions but everything after the first sentence leads the reader to think this is a methods paper (i.e. by looking for chicks with PFL you can tell what is going on in the environment). I think the logic should be reversed since PFL is rare. Basically reword paragraph.
• Line 72-73: Reword . Do poultry get PBFD? Makes it seem like entire sentence is only referring to poultry. Something like…Feather loss can be caused by starvation, nutritional deficiences, pecking, shock molt, and PBFD. Feather loss is rare in the wild and more common in poultry.
• Line 82: Why is it a conservation concern if it is a re-emergence 30 years later in a few chicks both times and went away? Could it be that it is a conservation concern that PFL was not seen in all of these long-term studies until recently and that more and more species keep exhibiting similar symptoms?

Materials & Methods
• Were you previously checking PFL chicks prior to them losing feathers? With what accuracy do you know their ages?
• Line 92-94: Confusing. How many chicks were banded each year? This will give us an indication of prevalence. Did you have any PFL in study nests?
• Line 94: How accurate do you know hatching date (i.e. age)?
• Line 94: How does near-daily in 2014 differ from 1-4 days in all other years? Be specific.
• Line 95: We stopped checking the nests every 1-4 days on 24 July when the majority of chicks were x days old…
• Line 96: How many individual chicks did you discover with PFL?
• Line 96: Add in that PFL chicks were study chicks and on all previous checks exhibited normal conditions.
• Line 96-97: Specify that you were measuring PFL chick feathers as they grew back in, not as they were still falling out.
• Line 99: How many normal chicks did you measure?
• Line 100: Where did the decision to test Corticorsterone come from? Need to set up in introduction what it can tell you about a developing chick.
• Line 100,103: What is this period?
• Line 101: Remove “most” and say sample size
• Line 103: What did you hope to get from checking cort levels in newly grown feathers? It seems like cort levels in feathers would only be important if the disorder occurred during feather development. What feathers were lost from PFL: down or secondary covert feather? What feathers grew back in? At what time do chicks usually do this transition? Did PFL occur at this time? If so, maybe the newly emerging feathers could have some indication of the problem. Need background in how feathers grow in terns. Did the normal chicks also change feathers during that time.
• Line 109: What was extracted from the feathers using methanol?
• Line 116: Be specific when possible. The larger sample size is 159.
• Line 117-119: Used linear growth rate for wing and tail. Do these measures grow linearly? Many species follow a period of slow growth early and late in the growth period and depending on when you took measurements, you’re linear growth rate may not accurately represent the data.
• Line 118: Mann-Whitney U Test is a nonparametric test. Why was this used? Are your sample sizes large enough to run a test. Does the median of 3 values accurately describe the population?
• Line 120: Why do you only include 3 control chicks?
• Line 123: 1 May to 31 July of what year?

Results
• Need to know how many chicks you looked at in total.
• Line 133: Sentence states that the condition is similar to territorial aggression but doesn’t show other symptoms, but in the notes on PFL datasheet, 6 out of the 8 PFL chicks have some note about being plucked/pecked. Can you explain…
• Line 138: Reword. Something like…Three chicks were found between 9 and 11 July and one chick on 24 July.
• Line 141: Remove “many”. Be specific. How many chicks had siblings?
• Line 143: How long was the chick dead at time of necropsy?
• Line 147: What is high compared to? Was the cellularity higher than normal tern chicks or higher in general than other species? Higher than adult terns?
• Line 150: Include stats (p-value).
• Line 152: Include sample size.
• Line 156: Reword. Something like…Median wing and tail growth was similar for normally-feathered (STATS) and PFL chicks (STATS).
• Line 163: Include stats.
• Line 161-170: Need to clarify when and what measurements were taken. Were all chicks weighed and measured on their last date seen? Was this last date seen close to fledging age? Line 162 says that it was but Line 164 says that it wasn’t.
• Line 164-167: Why are you projecting fledging ages when you know the age of the chick and when it was last seen? How do you know actual fledge dates of normally-feathered chicks? Why was the chick last seen on that date? Was it because you stopped checking nests or because when you returned the chick was gone? I would prefer to know what you saw rather than projections. So each chick has a range of fledge dates. If checked nest when chick was 30 days old and again 7 days later then you can say it fledged when it was between 30-37 days of age. Which is 5-12 days longer than normally-feathered chicks.
• Line 169-170: How did you estimate this? What does this number tell us? What was the colony size prior to this date?
• Line 171-177: Seems like a stretch. Need to include data from other years? Has this ever happened before? If it has, why didn’t chicks have PFL in those cases? Mean temperatures should be highly correlated as they include those min (night time) and max (daytime) temperatures. What is the purpose of this testing?

Discussion
• Line 186-187: If PFL and normally-feathered chicks have similar feather growth rates and at fledging all PFL chicks should have same wing length as normally-feathered chicks, what do you mean by the PFL chicks had shorter wing and tail feathers if you don’t know when they actually fledged? Were you comparing at a specific age. Reword. Something like “…had shorter wings and tails than normally-feathered chicks at age 24-27 days…”
• Line 191-210: Do you think you should be comparing your PFL chicks to those that had other associated symptoms?
• Line 213-216: In 2014 you don’t check nests near-daily later in the season, so your point that you would catch all dead PFL chicks doesn’t work unless your PFL chicks left before July 24th.
• Line 232-234: But they were similar (p=0.36)
• Line 235: Isn’t it just a reflection of the cort level while the feather was being grown? “…accumulation of cort during the growth of that feather”. Need to make the point that feathers don’t continue to accumulate cort after they are fully developed.
• Line 238: Change to “Other reports of PFL in terns…”
• Line 247: “…did not see much evidence…” this impies that you saw some evidence. What was it? Why is C. botulinum important here? Does it cause feather loss? Have these other species had C. Botulinum before?
• Line 251: Remove “historical”
• Line 251-254: If research has been debunked probably better not to include it
• Line 258-261: If the weather event is to blame, then you are correct that it is not a bioaccumulation event. But it could still be caused by the weather bringing up lots of contaminants and the birds being exposed all at once to those contaminants or it could be the weather had nothing to do with it and there is a bioaccumulation of pollutants in the chicks or their parents. Need to clarify what you are saying here.
• Line 262: Reword. Something like…Exposure to unknown toxins or pathogens seems the most likely cause of PFL…
• Line 264: Change “…1-2 weeks of age during this time period…” to “…1-2 weeks of age at the onset of PFL…”
• Line 288: Not sure reemergence should be used when other symptoms are different. For example, van Heezik and Seddon 1992 claimed that PFL was caused by being drastically underweight while the more recently affected penguins are not under weight.

Table
• Magellanic Penguin years should be 2007-08
• African Penguin years should be 2006-08

Figure 1
• Remove “of” from “…resulting from of premature…”
• Change “versus” to “and”

Figure 2
• Use box and whisker plot instead of bar graph.
• PFL sample size n=7 conflicts with that reported in the materials and methods on Line 101

Figure 3
• Need to show average line for each group as well. So that the reader can see that they begin differing at 10 days of age which you state on Line 155.
• Change “…for normal chicks (n..” to “…for normal chicks in 2014 (n…”

Figure 4
• Gray box should highlight the date range of when chicks began losing feathers. Unclear why you would use water surface temp to compare between all graphs.
• Fix y-axis of air temperature graph.
• Air temperatures are too high to be ‘C and to low to be F.
• Remove “compass” from y-axis label. Just say degrees.

Reviewer 2 ·

Basic reporting

The title is rather long and I’m not sure what “contemporary” refers to. Also “in” Lake Ontario sounds like the terns breed in the water.

This is a condition that has only been reported in the literature a few times and terminology is not standardized. The authors use the term “premature feather loss” and the acronym “PFL”. Although it’s convenient to abbreviate the term, we really don’t need another acronym that’s not obvious. Also, what does “premature” refer too? Why is it premature? The molt timing of common tern chicks should be explained and compared to chicks with feather loss.

Experimental design

Lines 234-237: The cort in feathers accumulates during feather growth. If the stress associated with the loss of feathers was short lived, sampling regrowing feathers wouldn’t detect it.

Validity of the findings

Measuring corticosterone in feathers being regrown doesn’t address the cause of the feather loss. That would require sampling feathers that were lost, but only if they were still growing when the stress occurred. The lack of a significant difference between feathers of chicks with and without feather loss indicates that the chicks were not overly stressed by feather loss.
Most of the feather samples that were collected for corticosterone were feathers that were being regrown after feather loss. Only one chick had old feathers sampled. It wasn’t clear from the spreadsheet in supplemental data which cort measurement was the old feathers. I assume it’s the row labeled “2014-FO”, which is by far the highest value in the spreadsheet. It’s an n of 1 so no conclusions can be drawn, but it should not be included with the other samples, which are regrowing feathers. It could be discussed in the context of pre- versus post-feather loss stress.

The connection to environmental variables and possible toxins is highly speculative. It’s intriguing that the feather loss coincided with unusual temperature and wind conditions, but how unusual were the conditions? Does something similar occur every summer or most summers, or only every 25 or 100 years? If it occurs nearly annually, but feather loss does not, that would argue against any connection.

Additional comments

It’s important to get reports of this condition into the literature, but this manuscript needs some work before it’s publishable.
Line 215: “seemly” should be “seemingly”
Lines 223-224: How much does a coat of feathers weigh? There should be some measurements of similar-sized birds in the literature that would be useful.
Line 239: Median hatching date is more meaningful than mean date if a few chicks hatch very early or very late.
Line 242: I don’t see a decrease of 13° C in Fig. 4. It looks like ~10° in water temperature. The axis labels are wrong for air temperature, and it’s not clear what it’s supposed to be.
Table 1: Is this only wild populations? If so, insert “wild” before “waterbirds” in the caption.
Fig. 2: “pc/mm” should be “pg/mm” in the caption. Line 101 says 6 chicks were sampled, but the caption here says n = 7. Which is correct?
Fig. 4B: Air temperature is 99-101+° C? Please correct.
Fig. 4D: Please label y-axis with N, E, S, W, & N, or 360, 90, 180, 270, & 360°.

Reviewer 3 ·

Basic reporting

No comments

Experimental design

The sample size in the corticosterone analysis portion of the article was small and may influence the results.
Feather extraction methods: Was mass recorded for each feather measured? Hayward et al 2010 and Lattin et al 2011 showed significant mass effects if too little mass extracted (inflated values if mass below 20mg). Important to note even if not using mass to report hormone values since this could affect ability to compare values if not taken into account.
EIA methods: Why was the antibody used here chosen (Assay Designs), rather than the one used in Bortolotti? Lattin et al 2011 warns that the antibody used can be important.
Were laboratory method validations performed for the corticosterone EIA, such as a parallelism/dilution series and an accuracy/spiked recovery test? Both of these validation steps are important ones to take to ensure that the antibody in the assay is recognizing the cort in a predictable manner, as well as confirming a lack of interfering substances. I noticed that the methods paper cited here, Bortolotti et al 2008, only performed parallelism and reported that a dilution series of the samples being parallel to the standard curve suggests lack of interfering substances. This statement is partially correct, since lack of parallelism would indicate a strong interference and/or low antibody affinity, but while parallelism can look great, interference can be detected at certain dilutions when performing an accuracy test (also known as a spiked recovery test, where you “spike” the known hormone standards with your sample and measure the amount “recovered”).

Validity of the findings

See above for lab method questions. It is unclear to me whether appropriate lab validation steps were performed which could influence the validity of the corticosterone findings. Since cort levels are being reported here as a potential cause for PFL, it will be important to confirm that the methods are being performed correctly.

Additional comments

Why no mention of the use of fecal samples here? Only blood (and feathers) are mentioned as methods of cort detection. It should at least be introduced, since it represents a pool of several hours up to days depending on the species, similar to feathers but not quite as long of a time period.
Typo in cort figure legend (pc/mm instead of pg/mm)

---

## Round 0.2 · Minor Revisions

This revision is much improved. I would suggest that you play down the temporal association more than you have. Your explanation is highly speculative and you can not rule out other causes including disease or toxins. Reduce the discussion because of lack of evidence to support your claims. I'd suggest removing line 209 to 217. Because you have no evidence drop line 237 after shown in Fig 3). to line 255 and pick up again on line 255 at "The extent of lake-surface... Drop line 258 through 261.

Drop on line 265 "Thus, it is possible to 276 and end with..." We favor exposure to unknown toxins or pathogenic organisms as the causal mechanism"

Re-organizing the abstract would help as follows

Take out sentence"There was striking temporal association" and move it. The abstract would continue on with "We found no..." and place the sentence "There was striking..." here then back to "PFL was not observed among common terns at Gull Island in 2015 but we saw similar feather loss in a herring gull () chick PFL is now known for gulls, penguins and now terns.

---

## Round 0.3 · accepted · Accept

In my opinion the manuscript is Acceptable for publication.